# A Review on the Management of Symptoms in Patients with Incurable Cancer

**DOI:** 10.3390/curroncol32080433

**Published:** 2025-07-31

**Authors:** Florbela Gonçalves, Margarida Gaudêncio, Ana Rocha, Ivo Paiva, Francisca Rego, Rui Nunes

**Affiliations:** 1Internal Medicine and Palliative Care Service, Portuguese Institute of Oncology Francisco Gentil Coimbra, 3000-075 Coimbra, Portugal; 4196@ipocoimbra.min-saude.pt; 2Faculty of Medicine, University of Porto, 4200-219 Porto, Portugal; mfrego@med.up.pt (F.R.); ruinunes@med.up.pt (R.N.); 3Health Sciences Research Unit: Nursing, Nursing School of Coimbra, 3000-075 Coimbra, Portugal; anamnrocha@esenf.pt (A.R.); ivopaiva@esenfc.pt (I.P.)

**Keywords:** dyspnea, nausea, pain, palliative care, palliative medicine, psychological distress, quality of life, symptom assessment, symptom burden, vomiting

## Abstract

Symptomatic control is one of the main principles of palliative care. Its correct approach includes continuous monitoring of symptom progression and the implementation of pharmacological and non-pharmacological therapies. Updates on the relevance of cannabinoid use for certain symptoms and the applicability and reasonableness of palliative sedation are some of the topics to be addressed. In this review, the authors summarize the management of common symptoms in patients in palliative oncology care, as well as present a brief reflection on quality of life in this context.

## 1. Introduction

With the aging population and increasing survival rates, chronic diseases such as cancer, often diagnosed at advanced stages of the disease, are increasingly prevalent, as are palliative needs [1]. Cancer patients receiving palliative care tend to have many difficult-to-control symptoms, such as pain, fatigue, dyspnea, anxiety, and depression, requiring the attention of a multidisciplinary team highly trained in end-of-life management [1].

If the exponentially increasing need for palliative care teams cannot be met by available human resources, the situation is at risk of becoming a public health crisis [2].

Palliative care consists of a multidisciplinary approach whose objective is to alleviate suffering and prioritize the quality of life of patients facing serious and life-threatening diseases [3]. Palliative care differs from end-of-life care in that the former can begin at any stage in the trajectory of a life-threatening illness, including the diagnosis, while the latter is intended for patients whose estimated prognosis is less than six months of life [4].

Palliative care is based on a series of basic principles [5]: Care objectives should be defined on the basis of the known disease trajectory, the degree of reversibility, and the patient’s values [6]. Symptom control should be attempted early and be intensive. Working in palliative care means working as a team [6]. Palliative care neither accelerates nor postpones death [6]. Finally, the patient’s family should always be involved in care [6].

Immunotherapy has changed the traditional trajectory of cancer, leading to greater success in many incurable tumors, but also to greater unpredictability [7]. The survival of patients undergoing immunotherapy can be longer in those who are responders, and even after its suspension, the periods of remission are also longer. Palliative care should support these new anti-neoplastic therapies by providing support in the control of physical, emotional, social, and spiritual symptoms [7].

Effective symptom control is a basic pillar of palliative care [7]. Pain, nausea and vomiting, and dyspnea are the focus of symptomatic assessment in palliative care, but patients experience other equally important symptoms that do not receive as much attention and are often overlooked, which negatively impacts the quality of life of these patients [7]. This may be because the patient does not mention them or because they are difficult to address by the team [7].

A symptom can be defined as a “subjective experience resulting from changes in the biopsychosocial, sensory, or cognitive profile of an individual” [8].

The proper management of pain and other symptoms is an essential component of palliative care [9,10], with the aim of maximizing comfort and quality of life for patients and their families [9,10]. The performance of palliative care is associated with a lower symptomatic burden and less suffering [9,10].

## 2. Methodology

In this work, the authors intend to present a narrative review article on approaches to managing the main symptoms of patients in oncology and palliative care while also emphasizing the importance of their impact on quality of life.

This article presents a brief description of the main symptoms in advanced cancer, aimed at oncologists, palliative care specialists, and other healthcare professionals who deal with patients with incurable cancer.

The authors conducted a bibliographical research in the PUBMED, CINAHL, and Web of Science databases (literature over the last 10 years), complemented by books and websites on the topic presented. The research was based on the following keywords: “Dyspnea”, “Nausea”, “Pain”, “Palliative care”, “Palliative medicine”, “Psychological distress”, “Quality of life”, “Symptom assessment”, “Symptom burden”, “Vomiting”. However, the authors decided to keep some essential articles on this topic, exceptionally, regardless of the publication date.

After the search was complete and all duplicates were thrown out, the authors reviewed the abstracts and articles to ensure that they addressed the objectives. So, the articles were chosen in order to develop each topic.

## 3. Symptom Management

### 3.1. Principles of Symptom Management in Oncology and Palliative Care

The symptoms of patients in oncology and palliative care are generally the same regardless of the diagnosis, especially at the end of life [11]. Most patients in these settings have multiple symptoms, and although they are traditionally addressed in isolation, more recent studies show a tendency to evaluate them as symptom groups [11]. An opportunity to optimize symptom control is to analyze clusters of symptoms described by the patient [11].

The literature shows that physical and psychological symptoms occur simultaneously [12]. For example, improving dyspnea reduces tiredness but also sadness and existential anguish [12,13].

Symptom control impacts the quality of life of patients in palliative care, so it is imperative that symptoms be accurately evaluated by either the patient or the caregiver when the patient is unable to do so [8]. Physical symptoms are better scored by the family than psychological symptoms, given their subjective nature [8].

Symptoms have several dimensions, including intensity, frequency, the degree of interference with basic activities of daily living, and the degree of associated suffering [8]. Some symptoms, such as dyspnea, can even be predictors of prognosis and survival [8].

Correctly and continuously monitoring the severity of symptoms, as well as the possible appearance of others, is essential for effective symptom control and for making necessary therapeutic adjustments [8]. To this end, an interdisciplinary team is needed that, in sum, has the ability to listen, monitor, and develop a chronic disease management strategy that minimizes pain and suffering [8,9].

In palliative care, evaluating symptoms is essential for making the approach to their management as effective as possible [14]. When assessing symptoms, it is essential to obtain the classification of their intensity over a certain period of time, such as the last 24 h [14]. Such is the case for the Edmonton Symptom Assessment System (ESAS), which evaluates the intensity of the nine main symptoms in patients receiving palliative care from 0 to 10 [14].

The ESAS-r is an adaptation of the original version of the visual analog scale ESAS, developed in 1991 in Edmonton with the aim of assessing the most common symptoms in palliative care [15]. Subsequently, it was classified numerically with a 24-h assessment period [15]. The ESAS-r was created in 2011, and the assessment period changed from 24 h to now [15].

Another instrument widely used in palliative care to measure the symptomatic burden is the Palliative Outcome Scale (POS), which evaluates the intensity of pain and other symptoms experienced in the last 3 days on a Likert scale scored from 0 to 4 [16]. This is a more comprehensive scale, since it has questions that also allow the psychological, social, and spiritual dimensions to be measured, and it is often used as an instrument to assess the quality of life and the care and well-being of patients in palliative care [16].

Some patients are unable to rate the severity of their symptoms on a numerical scale because of poor health literacy or altered cognitive status [17]. An alternative is to ask patients how they rate their symptom distress rather than how intense the symptoms are and provide them with a range of possible responses (e.g., not at all, a little, a lot, or very much) [18]. Some symptoms, such as fatigue, may be rated by patients as mild in intensity but severe in suffering/distress [18]. Information on symptom duration, nocturnal awakening, and interference with activities of daily living can also help assess symptom severity [18].

Thus, it is important to conduct a careful evaluation centered on the patient and family and identify triggers that may interfere with the quality of life of both. Palliative care gives humanity and dignity to the therapeutic approach to these symptoms.

### 3.2. Pain

Pain is defined as “an unpleasant sensory and emotional experience associated with actual or potential tissue damage”. This concept was revised by the International Association for the Study of Pain (IASP) in 2020 [19]. Pain is a personal experience influenced by the physical, psychological, emotional, social, and spiritual factors of the affected person and by their life experiences [17,19].

Recently, increasing importance has been given to the concept of “Total Pain”, first described by Cicely Saunders in 1964, which refers to the pain felt by patients with life-threatening diseases who do not respond to standard analgesic therapy, given its multidimensional character (physical and non-physical dimensions) (Figure 1) [20,21]. It is known that emotional pain is a form of spiritual coping, in which hope is associated more with the psycho-emotional component of pain than with its intensity [20,21]. The concept of “Total Pain”, created several decades ago, remains the gold standard concept for a symptoms approach in palliative care.

Chronic pain negatively affects the quality of life of cancer patients [22]. Its prevalence is high, being present in about 39.3% of cancer survivors, in 55% of patients undergoing curative treatment of the disease, and in 66.4% of those with advanced, metastatic, or terminal cancer [22]. In more than one-third of patients (38%), the score given was >5 on the numerical pain scale, meaning they experienced moderate to severe pain [22].

Although the prevalence of cancer is increasing worldwide, it is also true that the survival of patients is longer thanks to new therapies [23]. Pain is the symptom that affects cancer patients the most, which can tend towards chronicity [23]. Despite its high prevalence, chronic pain is not described in the current International Classification of Diseases (ICD-10) [23]. Cancer-related chronic pain is characterized by its association with cancer or its treatment [23]. The new classification of chronic oncological pain (lasting more than three months) in the ICD-11 aims to assist in the individualization of care and in the understanding of these painful syndromes [23].

Since pain is considered the 5th vital sign, its assessment should be a routine part of daily clinical practice [24]. The evaluation, treatment, and monitoring of pain should be conducted whenever there is a recurrence of cancer or a second neoplasm appears [24]. The initial approach to oncological chronic pain involves the use of non-opioid analgesics or adjuvants, whether for symptomatic relief or for improvement of functionality [24]. Whenever this strategy is ineffective and there is suffering, opioid analgesics should be carefully considered in cancer patients, taking into account the potential appearance of adverse effects [24]. Healthcare professionals should be familiar with terms such as tolerance, dependence, and abuse, and adopt measures that minimize the potential for addiction [24].

The correct approach to managing pain involves evaluating and measuring it [20,25]. There are several measurement tools, such as the Brief Pain Inventory, the Palliative Outcomes Scale (POS), the Edmonton Symptom Assessment System (ESAS), and the Visual Analog Scale (VAS) [20,25]. Visual Analogic Scale (VAS) can be used carefully in patients with cognitive impairments [26]. For example, a review showed that the VAS is used in about 16% of studies addressing pain assessment in dementia and that the results are unreliable when cognitive impairment is moderate to severe [26].

Individuals with dementia or cognitive impairment may not be able to verbally convey the level of pain they are experiencing, leading to undertreatment [27]. For this reason, the literature suggests using the Pain Assessment in Advanced Dementia (PAINAD) scale [27].

In the oncological context, pain may be caused by the direct effect of the neoplasm (invasion/progression to viscera, bone, and/or soft tissue, raised intracranial pressure, nerve injury, or a combination of these) [28]. On the other hand, some pain may be related to the side effects of oncological treatment (e.g., nerve injury during surgery, chemotherapy-induced peripheral neuropathy, lymphoedema, muscle spasms, pressure ulcers, or constipation) [28].

Thus, cancer pain can be classified as nociceptive (visceral and somatic), neuropathic (nerve compression, nerve injury, and sympathetically maintained), or mixed. Its correct identification is essential in analgesic guidance [28].

Pain treatment with drugs alone is insufficient. Non-pharmacological measures are also needed to target the multiple dimensions of pain as the “total” symptom that it is [20]. A correct pharmacological approach to managing cancer pain consists of multimodal analgesia, in which several classes of analgesics are combined to reduce adverse effects and control pain more effectively [29,30].

Therefore, in addition to pharmacological interventions, cancer pain treatment must be based on an approach that includes social, spiritual, psychological, neurosurgical, radiotherapeutic, physiotherapeutic, and anesthetic interventions [28].

In 1986, the World Health Organization (WHO) first published the three-step analgesic ladder, in which drugs and the potency of analgesic drugs accompanied the severity of the pain (Figure 2) [28,29,30]. Recent studies suggest an analgesic ladder of only two steps, in which the second is eliminated [31,32].

When controlling pain, it is important to follow some basic principles [28]. Whenever possible, the oral route should be used because it is the least invasive (“by mouth”); the schedule should be regular (“by clock”); medications should be administered at the lowest effective dose; and those with the least adverse effects should be chosen (“for the individual”; “with attention to detail”) [28]. It is important to pay attention to the details and particularities of each patient [28].

Opioids still remain the gold standard of treatment for moderate to severe cancer pain [22]. Consequently, the appropriate use of opioids is very important for effective pain relief, taking into account the benefits and risks of these drugs [28,29]. Opioids are safe and potent drugs with important side effects [28,29] and can be administered by almost any route [28,29]. When choosing an opioid, we must take into account the individual characteristics of each patient and the associated comorbidities [28,29].

Weak opioids (such as tramadol and codeine) are recommended for initiation when pain intensity is reported to be mild or moderate, particularly in opioid-naïve patients [28,29,33,34,35,36]. These drugs are usually combined with acetaminophen/paracetamol and/or non-steroidal anti-inflammatory drugs (NSAIDs) [28,29,33,34,35,36].

**Figure 2 curroncol-32-00433-f002:**
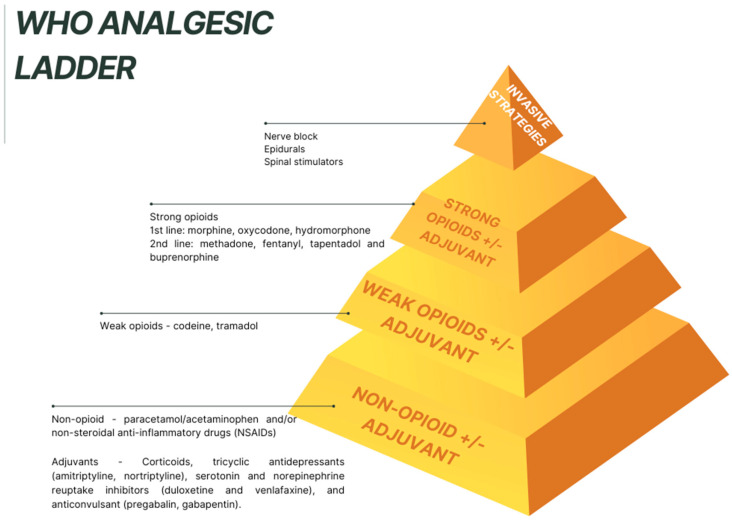
WHO analgesic ladder for cancer pain management (with suggested 4th step). Based on WHO Guidelines and Vargas-Schaffer et al. [37].

On the other hand, strong opioids (first line: morphine, hydromorphone, and oxycodone; second line: methadone, buprenorphine, and fentanyl) are recommended when pain intensity is moderate or severe [28,29,33,34,35,36]. It is important to start with a low dose and titrate up in order to ensure adequate analgesia and tolerable side effects [28,29,33,34,35,36].

No studies have shown that one opioid is superior to another, and morphine continues to be the most prescribed opioid because it is the best known, effective, and low-cost [28,29,33,34,35,36]. Fentanyl and buprenorphine are reserved for cases where renal failure is present, especially when creatinine clearance is less than 30 [22,29,30,31]. Fast-acting formulations, such as fentanyl administered sublingually, transmucosally, or nasally, are excellent drugs for breakthrough pain [28,29,33,34,35,36].

Approximately 26% of patients respond poorly or not at all to opioids [37,38,39]. There are many possible causes of this phenomenon, including breakthrough pain, disease/cancer progression, negative psychological conditions, and the presence of a neuropathic mechanism [37,38,39].

About 20–40% of cancer patients will experience neuropathic pain, defined by the IASP as pain “caused by a lesion or disease of the somatosensory nervous system” [40]. In these patients, the nervous system (peripheral or central) may be affected by the tumor itself or by its treatment (chemotherapy, surgery, and radiotherapy) [40,41].

The NeuPSIG (Neuropathic Pain Special Interest Group) grading system was updated in 2016 with the aim of improving the degree of certainty that pain is neuropathic in nature, classifying it into three levels: possible (compatible clinical history and location), probable (when associated with neurological dysfunction), and definite (when complementary diagnostic tests confirm the presence of somatosensory injury) [42].

The medications used to treat neuropathic pain include anticonvulsants (pregabalin, gabapentin), tricyclic antidepressants (nortriptyline, amitriptyline), and serotonin and norepinephrine reuptake inhibitors (venlafaxine, duloxetine) [43]. These drugs have also been called adjuvant medications because they are used to treat nociceptive pain (with an additive effect when used in combination with opioids and non-opioids) [44].

As with opioids, clinicians must balance the benefits and adverse effects of adjuvant medications [45,46]. Tricyclic antidepressants have anticholinergic properties and can induce blurred vision, dry mouth, sedation, dry urinary retention, orthostatic hypotension, and tachycardia [45,46]. On the other hand, serotonin and norepinephrine reuptake inhibitors can induce headaches, dizziness, nausea, sweating, and arterial hypertension [45,46]. Finally, patients on anticonvulsants can experience somnolence and dizziness and are at risk of respiratory depression when taking opioids as well [45,46].

Cannabinoids have increasingly gained attention [47,48]. However, their use in pain management is still controversial [47,48]. The legalization of medical cannabinoids in 40 countries has increased interest in their use in cancer symptom management, especially in pain control [47,48,49]. In fact, cancer patients use cannabis regularly [50]. A study carried out in Canada revealed that 18% of cancer patients admitted to having used cannabis in the last 6 months, with 46% of these doing so to control pain [50]. In fact, most studies conducted on cancer patients in palliative care do not show that adding cannabis to opioids reduces pain [48,51].

In the literature, there are several proposed updated versions of the WHO analgesic ladder [37,52,53]. In these new versions, the concept of integrative medicine (which can be implemented at every step) is proposed, along with a fourth step, which includes interventional/invasive treatments [37,52,53].

Interventional/invasive treatments may be considered at any point in the cancer trajectory but may be proposed particularly when patients have uncontrolled pain and/or analgesics have “intolerable adverse effects” [44]. The two main categories are intrathecal/epidural analgesics and nerve blocks [44]. Intrathecal/epidural procedures can be used to simultaneously administer different classes of analgesic drugs, as well as allow the use of medications that are not available in oral/transdermal formulations [44]. Deciding whether to use these interventions requires an assessment of prognosis, risks, contraindications, and benefits [44]. Nerve blocks are traditionally considered the last step in the analgesic ladder; however, evidence has shown their action to be more effective when applied in the early stages of cancer [44].

Finally, more recently, the European Society of Medical Oncology (ESMO) recommended an integrative strategy for the treatment of pain in cancer patients [32]. The suggested strategy included an approach based on disease-directed treatments (primary anticancer treatments such as surgery, hormonotherapy, chemotherapy, and radiotherapy) and classical and interventional analgesic therapies, combined with a variety of non-invasive and non-pharmacological therapies [32]. Therefore, it is important to highlight the role of non-pharmacological approaches that are part of the integrative medicine strategy proposed for pain relief [54,55]. The objective of these strategies is to reinforce the intervention of pharmacological measures and improve the comfort and quality of life of patients [54,55]. To this end, several approaches have been developed, including tai chi, hypnosis, meditation, yoga, mindfulness, cognitive behavioral strategies and pain coping, music therapy, acupuncture, and massage therapy [54,55]. These therapies have demonstrated benefits in controlling anxiety, pain, depression, and fatigue and providing a sense of well-being through a personalized strategy according to patients’ preferences [54,55].

Pain management in the context of oncology faces daily challenges, which are more evident in the advanced stages of the cancer trajectory, highlighting the importance of a multidisciplinary team with the aim of minimizing the impact of this symptom on the patient’s life.

### 3.3. Dyspnea

Dyspnea is defined by the American Thoracic Society as “a subjective experience of breathing discomfort that consists of qualitatively distinct sensations that vary in intensity” [56,57].

Thus, dyspnea is a complex symptom that potentially poses a critical threat to homeostasis and, therefore, often prompts adaptive responses [57].

Dyspnea can be potentiated by other symptoms present in cancer patients in palliative care, such as fatigue, depression, and anxiety, limiting their quality of life and causing great apprehension to family members [57,58,59]. The presence of dyspnea is a marker of poor prognosis [57,58,59]. Prolonged or intractable dyspnea causes distress and suffering and compromises performance and quality of life [57].

It is important to emphasize that physiological measures (e.g., heart rate, oxygen saturation, and respiratory rate) have little relationship to the patient’s experience of dyspnea [58,59,60,61,62,63]. The patient’s subjective rating remains the standard for assessing symptom burden, notably when evaluating the benefits of any therapeutic strategies provided [58,59,60,61,62,63].

The interaction between dyspnea and other symptoms highlights the multidimensional nature of the distress and suffering that a patient experiences [58,59,60,61,62,63].

Dyspnea is present in 10 to 70% of patients with advanced cancer. The approach to its treatment should be performed in a hierarchical manner, starting with the investigation of potentially reversible causes whose reversal does not have disproportionate adverse effects on the patient’s well-being [58,59,60,61,62,63].

Pharmacological treatment is multimodal. In terms of drugs, corticosteroids and bronchodilators may be indicated, as well as antibiotics, whose objective is symptom control [63,64]. Studies on the possible benefits of inhaled furosemide, antidepressants, or neuroleptics have produced conflicting results [63,64].

There may be a need to move on to respiratory sedatives for dyspnea control, such as opioids (morphine, methadone, fentanyl, oxycodone, or codeine) and benzodiazepines (midazolam, diazepam, or lorazepam), the latter of which are used when there is associated anxiety or a need to increase doses of opioids [63,64]. Continuous palliative sedation should be considered in patients who have dyspnea refractory to standard treatment and whose life expectancy is short [64].

In addition to pharmacological treatment, the palliative care team will offer non-pharmacological measures, such as a fan or an open window, and pharmacological measures that include oxygen supplementation to increase comfort [64].

Patients with dyspnea often report that cool air movement reduces dyspnea/breathlessness, and studies have shown that air directed at the face decreases induced dyspnea. For this purpose, the use of fans and/or cold airflow for the relief of dyspnea is suggested (the air movement across the face stimulates the trigeminal nerve) [57]. Exercise and pulmonary rehabilitation programs may reduce dyspnea scores and improve quality of life [60].

High-flow oxygen therapy and non-invasive ventilation may be indicated at the end of life to control dyspnea or to prolong life for short-term goals, such as the arrival of a family member or to allow a farewell [60,65].

In any case, the use of oxygen therapy as a palliative measure is sometimes controversial.

Oxygen is a therapeutic measure in clear clinical situations, and therefore, oxygen therapy must be prescribed and appropriately administered [65]. The administration of oxygen through the various available interfaces (nasal cannula, face mask, high-flow mask) is based on the principle that increasing the fraction of inspired oxygen (FiO_2_) will increase alveolar oxygen pressure and, consequently, will increase arterial oxygen pressure, improving tissue oxygenation [65].

However, there are some risks associated with the use of oxygen, namely, cerebral and coronary vasoconstriction, hypercapnia, increased peripheral vascular resistance, pulmonary atelectasis, the production of reactive oxygen species with cytotoxic effects, decreased cardiac output, discomfort, and airway damage [65].

Over the years, researchers have hypothesized that oxygen therapy has a beneficial effect in the management of dyspnea in patients receiving palliative care [65]. Its systematic administration to these patients was based on a personal intuition that it could be beneficial and on a certain belief that, in view of such threatening conditions, it would have an almost miraculous power [65]. The possible symptomatic improvement would be due to contact of the face with fresh air, which would stimulate the trigeminal/vagal nerve and reduce dyspnea [65].

Thus, the use of oxygen therapy in palliative care should be reserved for patients with persistent peripheral saturation < 90% (after identifying and optimizing therapeutic interventions in the various possible causes) or for patients whose dyspnea significantly improves after a course of oxygen therapy [65,66]. In non-hypoxemic patients, non-pharmacological interventions and opioid drugs should be tried first, simultaneously [65,66]. Oxygen therapy should be discontinued in the absence of clinical benefit or when disadvantages (for example, marked discomfort caused by masks or nasal cannulas, dryness of mucous membranes) outweigh the benefits [65,66]. When all other strategies have been optimized, a therapeutic trial for non-hypoxemic patients can be considered, with a recommended duration of 72 h, with adequate and systematic clinical reassessment. In the absence of a benefit, this therapeutic trial must be suspended [65,66].

The effectiveness of the palliation of dyspnea depends on the availability of a specialized palliative care team that, with its knowledge in multiple fields, knows how to minimize the impact of this symptom on the quality of life of the patient and the patient’s family.

### 3.4. Nausea and Vomiting

Nausea and vomiting are symptoms that palliative care patients describe as very unpleasant and are present in about 68% of cancer patients [67]. In the last 6 weeks of life, nausea and vomiting are present in more than 40% of patients [67,68].

Uncontrolled nausea and vomiting often affect the cognitive and psychosocial dimensions of the patient, with a negative impact on their quality of life and contributing to a fear of death by hunger or thirst [69].

Nausea is the subjective sensation of vomiting accompanied by symptoms of dysautonomia, while vomiting is the expulsion of gastric contents through the oral cavity [70].

The literature has little evidence and information about the mechanisms involved in triggering nausea and vomiting in advanced cancer [70]. Thus, an etiological approach based on a detailed clinical history and objective examination is strongly recommended so that the most appropriate antiemetic is chosen based on the receptors implicated [70].

Nausea and vomiting are common and distressing symptoms affecting the majority of patients in oncology and palliative care settings [70,71]. Assessment and treatment have been based on understanding the physiopathology and neurotransmitters involved in this process [70,71].

In advanced metastatic cancer, the most common causes of nausea and vomiting are metabolic and hydroelectrolytic disorders, among which hypercalcemia, liver and kidney failure, opioids, and intestinal occlusion stand out [71].

Antiemetics should be administered at fixed times, with progressive dose titration, and preferably parenterally [72,73,74]. If ineffective, a combination of antiemetics should be attempted, given the multifactorial nature of these symptoms and the frequent involvement of several neurotransmitters [72,73,74]. Corticosteroids, olanzapine, and cannabinoids are promising drugs in the treatment of nausea and vomiting, although haloperidol and metoclopramide continue to be used [72,73,74].

Nutritional counseling and acupuncture are non-pharmacological measures that should be used concomitantly with prescribed drugs [72,73,74]. It is recommended to avoid environmental stimuli (such as smells, sounds, or sights) that may trigger nausea [75]. In terms of diet, spicy, fatty, and salted food should also be avoided [75]. Behavioral approaches, such as distraction and relaxation, can decrease psychological stress by redirecting the patient’s attention elsewhere and increasing the patient’s feelings of well-being [75].

In Table 1, the authors present an overview of nausea and vomiting mechanisms, the involved receptors, and the main drugs used according to etiology, because antiemetic drugs are predominately neurotransmitter-blocking agents [70,75].

### 3.5. Anorexia–Cachexia Syndrome

Cancer anorexia–cachexia syndrome (CACS) is a metabolic syndrome associated with oncological disease and “characterized by loss of muscle mass with or without loss of fat mass that occurs involuntarily” [76].

Anorexia–cachexia syndrome associated with advanced cancer is present in up to 80% of patients [77]. Its incidence ranges from 20 to 40% in early stages (diagnosis) and 70–80% in advanced cancer. The prevalences described in the literature, according to the primary tumor site, are approximately 80% (gastric and pancreatic neoplasms), 54–60% (prostate, lung, and colon cancers), and 32–48% (sarcoma, lymphomas, breast cancer, and leukemias) [76].

Anorexia–cachexia syndrome in cancer has been of interest to the scientific community in recent decades, resulting in the development of promising drugs and non-pharmacological measures [78]. However, no drugs have yet been approved for its treatment [78].

Anorexia–cachexia syndrome’s physiopathology is based on a combination of metabolic mechanisms and tumor anatomical and cancer treatment factors [76].

One of the main mechanisms is the excessive production and secretion of cancer-related cytokines, particularly tumor necrosis factor alpha (TNF-α), interleukin-6 (IL-6), interleukin-1 (IL-1), and C-reactive protein (CRP) [76]. These cytokines lead to a pro-inflammatory state that causes a catabolic and anorectic effect, resulting in lean mass and weight loss [76].

Most solid tumors, particularly gastrointestinal and head/neck neoplasms, can cause mechanical nutritional problems [76]. Gastric reflux, abdominal pain, and/or dysphagia may lead to a decline in food intake [76]. Some tumors can cause early satiety due to abdominal occupancy (e.g., hepatomegaly and ascites) [69]. Other cancers affect gastrointestinal motility (as a result of peritoneal and/or intestinal infiltration) and nutritional malabsorption (particularly in pancreatic cancer) [76].

On the other hand, oncological treatments can cause cachexia [76]. Surgery can alter swallowing, absorption, and/or digestive functions and thus lead to malnutrition [76], and chemo- and radiotherapy can lead to xerostomia, appetite loss, oral mucositis, nausea, vomiting, dysgeusia, and enteritis [76]. These effects make it difficult to eat and, as a result, cause a reduction in caloric intake, which exacerbates nutrient loss [76].

Three phases are recognized in anorexia–cachexia syndrome: pre-cachexia, cachexia, and refractory/advanced cachexia. The stage depends on the degree of weight loss and the reversibility of the condition with nutritional intake [79]. Pre-cachexia is characterized by systemic inflammation with weight loss of <5% of the patient’s body weight [76,80]. Patients in the cachexia stage present with systemic inflammation, weight loss of >5% (or BMI < 20 kg/m^2^ or weight loss > 2% if sarcopenic), and the loss of cell mass [76,80]. Finally, refractory or advanced cachexia is characterized by irreversible catabolism, resulting in poor functional status [69,73]. In this phase, the cancer also progresses, sometimes with an expected survival < 3 months, corresponding to the end-of-life phase [76,80].

Simple measures such as the Edmonton Symptom Rating Scale (ESAS) and the Subjective Patient-Generated Global Assessment (PG-SGA), together with nutritional assessment and a weight loss of more than 5% in the last 6 months, are used to identify patients with anorexia–cachexia syndrome [76,80].

The treatment of cancer cachexia is multimodal and includes a nutritional approach and pharmacological measures [76,80].

Nutritional support includes dietary counseling, particularly in patients who are able to eat [76]. This counseling consists of guiding the patient and/or family by promoting frequent and small meals according to the patient’s taste, with adjustments to the consistency, as well as choosing high-energy and high-protein foods, enriching their diet [76]. Nutrition plays an important role in the psychosocial well-being of advanced cancer patients, affecting their quality of life [81,82].

In addition to nutritional advice, patients can enrich their diet with the help of oral nutritional supplements [76,80]. These products are a balanced mixture of macro- and micronutrients available on the market in the form of liquid foods, puddings, and powder formulations that can be reconstituted with water or milk [76,80]. According to the literature and main guidelines in this area, these supplements are best used as adjuncts to the diet, because their use alone, without changes in the diet, is not effective [76,80]. Some studies have reported positive effects of using these supplements on energy intake, weight gain, and some aspects of quality of life (e.g., emotional functioning, appetite loss, and global quality of life) in cancer patients [76,80].

In patients expected to survive for more than several months and in those receiving anticancer therapy, nutrition should be intensified as needed through enteral and parenteral routes [76,80]. Enteral nutrition may be used in cases of dysphagia if the small bowel function is preserved (using tube feeding) [76,80]. Parenteral nutrition should be used if oral intake and tube feeding are not tolerated or considered inadequate [69,73]. Artificial nutrition is rarely indicated in palliative care, and prognosis and the possibility of complications should be taken into account [81,82].

The literature lacks sufficient evidence to support any drug to treat this syndrome [76,80]. Pharmacological treatment depends on the patient’s preferences but also on the degree of anorexia/weight loss, the risk of side effects, comorbidities, prognosis, and treatment goals [76,80]. The main benefits associated with pharmacological treatment are increased appetite and weight gain [76,80].

The two main important drugs used in the pharmacological treatment of anorexia–cachexia syndrome are corticosteroids and progesterone analogs (megestrol acetate) [82,83,84].

Corticoids improve appetite in a manner similar to that observed with progesterone analogs [76,80]. Given their adverse effects and the decline in their efficacy (with long-term use), their role is often limited to those with an estimated life expectancy of weeks to a few months [76,80]. In oncology and palliative care, the most commonly used corticosteroids are dexamethasone, prednisolone, and methylprednisolone [76,80]. No corticosteroid is superior to another, but dexamethasone is preferred due to its lower mineralocorticoid effects [76,80].

On the other hand, megestrol acetate improves appetite and body weight in cancer-related cachexia, but the weight gained is mainly in the form of adipose tissue [76,80]. Its possible complications include thromboembolic events, edema, and adrenal suppression, especially when the dosage exceeds 800 mg per day [76,80].

In the literature, other drugs for cachexia treatment are described, although the evidence regarding their benefits is inconclusive [76,80]. The drugs in question are olanzapine, anamorelin, thalinomide, omega-3 fatty acids, vitamins, minerals, cyproheptadine, non-steroidal anti-inflammatory drugs, mirtazapine, hydrazine sulfate, TNF inhibitors, melatonin, insulin, and cannabinoids [76,80].

Thus, the goal of cachexia treatment is to stabilize or increase protein–caloric ingestion, reduce inflammation, optimize the person’s nutritional status, and prevent negative effects related to malnutrition [76,80]. This goal is particularly important in the early stages of cancer and cachexia [76,80]. However, as the end of life approaches, the goal of care turns more toward quality of life than nutritional adequacy [85]. This is where the great dilemma arises between reconciling nutritional intervention with offering comfort and helping to control symptoms [85].

### 3.6. Psychological Distress

Optimizing the symptom control of patients in palliative care requires that the respective team be attentive to their emotional suffering [86].

Psychological distress is highly prevalent in the oncology context [87]. Although its identification, assessment, and management are supported by the literature, there is still no universal concept of distress and suffering [87]. For example, psychological distress could include anxiety, sadness, depression, fear, and negative affect [86].

According to the National Comprehensive Cancer Network (NCCN), distress in cancer is defined as “a multifactorial unpleasant experience of a psychological (cognitive, behavioral, emotional), social, spiritual, and/or physical nature that may interfere with one’s ability to cope effectively with cancer, its physical symptoms, and its treatment” [88].

In other words, psychological distress can be defined as the “unique discomfort and emotional state experienced by an individual in response to a specific stressor or demand that results in temporary or permanent harm caused to the person” [86]. Emotional suffering can encompass several dimensions, such as psychological, social, and spiritual [89].

The instruments that allow the measurement of psychological distress can be routinely used and include symptom assessment scales, as well as more specific tools and structured interviews for this purpose [90]. The Edmonton Symptom Assessment System (ESAS) and the Palliative Outcome Scale (POS) are generic instruments commonly used by palliative care teams to screen for symptoms, including anxiety, depression, well-being, meaning in life, and sharing feelings [90]. When the scores of these symptoms are high, patients should be referred for psychology or psychiatry consultations in which more specific instruments will be used, such as the Distress thermometer (DT), the Hospital Anxiety and Depression Scale (HADS), and the Brief Symptom Inventory (BSI) [90].

Depression and anxiety are the most common psychiatric disorders in palliative care [91]. It is estimated that 24 to 70% of these patients may experience depression, leading to a significant decline in their quality of life and even suicidal ideation [91].

Methylphenidate and antidepressants may also be beneficial in the treatment of depressive symptoms in this subpopulation of patients [92,93]. The combination of these drugs with non-pharmacological measures, such as music therapy, has been shown to be more effective than the use of pharmacological therapies alone [92,93].

Anxiety is very common in patients who are in palliative care and, like depression, has a strong impact on their quality of life. Anxiety is present in about 20% to 50% of patients with advanced cancer [93]. Some physical symptoms, such as nausea and anorexia, are more prevalent when patients do not feel well emotionally [94]. By reducing anxiety, the palliative care team is providing patients with holistic and humanized care that is attentive to the symptoms as a whole [94].

The correct approach to treating anxiety is an area that is still little studied and with little training [95]. The literature shows that screening scales are rarely used, and the use of benzodiazepines is often inadequate, being prescribed in patients with survival expectations of months [95]. Access to psychology and psychiatry consultations is also very deficient [95].

Cancer patients can benefit physically and psychologically from personalized distress interventions [96]. Patients who were routinely assessed for distress showed improvements in psychological and physical symptoms, as well as in well-being [96].

Nowels et al. conducted a systematic review that demonstrated the need to consider improvements in psychological symptoms as realistic outcomes of palliative care interventions [97]. However, non-worsening emotional symptoms may be a more reasonable goal for patients in end-of-life care [97]. This review further demonstrated that, for patients in the late stages of their disease trajectory, reducing psychological suffering was the most difficult, since distress tends to increase as the end of life approaches (especially in diseases with high symptom burden such as cancer) [97].

Paley et al. performed a systematic review that showed a wide range of therapeutic approaches to managing distress in cancer patients [98]. There was no consensus on any intervention method, although mindfulness was the most frequently researched with some evidence, followed by communication and group therapies [98].

Mindfulness appears to be effective for cancer patients, particularly those in the terminal stages of the disease [98]. This technique is easy to teach and can be self-administered outside medical settings by patients and/or caregivers [98]. They can be taught in person or online and practiced at home. Brief mindfulness may also be appropriate for use in end-of-life care [98].

Palliative care teams should make an effort to identify emotional distress so that patients can take advantage of targeted psychotherapeutic and pharmacological interventions and thus improve their quality of life.

### 3.7. Palliative Sedation

Throughout the disease process and especially in the last days of life, patients can present with symptoms that cannot be controlled, despite appropriate treatment, and may interfere with the process of a peaceful death [99]. As a result, some authors deem it a medical and moral imperative to use sedative drugs to control these symptoms [99].

According to the European Association of Palliative Care (EAPC), palliative sedation is a therapeutic measure of the last resort for the treatment of refractory symptoms. It entails the monitored use of sedative medication with the intention of inducing a state of decreased consciousness to alleviate otherwise intractable suffering in a manner acceptable to the patient, family, and health professionals [100].

A refractory symptom is a symptom that cannot be adequately controlled, despite exhaustive and tolerable pharmacological and non-pharmacological interventions that do not compromise the patient’s degree of consciousness [100]. The most common refractory symptoms are pain, dyspnea, anxiety, seizures, delirium, and psychomotor agitation, with an overall prevalence of 10–50% [93]. Refractory symptoms often increase as the patient gets closer to death and may interfere with a peaceful death [100].

The objective of palliative sedation is to control symptoms that cause severe discomfort and are refractory to conventional palliative treatment, relieving the suffering of patients in the final phase of the disease, improving their comfort, and maintaining the dignity of human life until its end [100].

According to the EAPC, sedation is used in palliative care within different contexts: to perform more invasive procedures, in end-of-life patients on ventilatory support, in the presence of refractory symptoms, in cases of emergent sedation, in respite sedation, and in cases of refractory psychological and existential suffering (2009 and 2023 recommendations) [100,101].

Emergent sedation is indicated in irreversible clinical situations in terminal patients who require immediate relief, such as catastrophic hemorrhages, asphyxia and suffocation, uncontrollable pain or dyspnea, and delirium with intense psychomotor agitation [100].

In the 2009 EAPC recommendations, there is no consensus on end-of-life sedation for severe non-physical symptoms (existential or psychological) such as depression, anxiety, demoralization, or existential suffering [100]. The lack of clear consensus on these issues is a barrier to the optimal treatment of patients with existential distress, particularly at the end of life, as well as a source of misunderstanding and controversy [102]. Many authors have raised concerns about the subjectivity and ambiguity inherent in the assessment of existential distress [102].

However, the 2023 EAPC guidelines recognize that existential distress is an indication for palliative sedation and includes a number of distinguishable non-physical components [101]. Existential suffering should only be considered refractory “after a comprehensive assessment by specialists in palliative care, considering the psychological, social and spiritual suffering” [101].

In summary, most guidelines and experts define the following criteria for palliative sedation: terminal illness (life expectancy of less than 6 months), imminent death (hours or days, maximum 2 weeks), intolerable suffering, refractory symptoms, the involvement of a palliative care specialist or an interdisciplinary team, and informed consent obtained from the patient or their legal representative [100,101].

This decision must be in accordance with the wishes of the patient, family, or legal representative and be the consensus of the medical team [100,101].

The level of sedation can be classified according to degree (light or deep) and duration (intermittent/respite sedation or continuous) [103]. When defining the level of sedation, there are some principles to consider: the level of sedation should always be the minimum necessary for the adequate relief of symptoms, and intermittent sedation should be the first option considered [103]. According to most authors, deep sedation should only be considered in the terminal phase of the disease, with an expected prognosis of hours at most [103].

The scale most commonly used to assess the state of consciousness/level of sedation is the Richmond Agitation Sedation Scale (RASS), which assesses the patient’s level of agitation and sedation, with a score ranging from +4 (combative patient) to −5 (unresponsive patient) [100,104]. It is an observational scale and validated for the intensive care context [100,104]. However, it has been modified and adapted to the context of palliative care and is being studied for its use in the monitoring of palliative sedation (Richmond Agitation Sedation Scale—Palliative Version (RASS-PAL)) [100,104].

The use of sedation to relieve the suffering of the patient can cause distress and stress among the family and health professionals involved in care [105]. One of the apparent reasons behind this stress is the fear that death is accelerated by palliative sedation [105]. In general, the evidence indicated that palliative sedation did not accelerate death, which was a concern of physicians when prescribing treatment, as well as of family members [105].

The ideal drugs for palliative sedation should have a rapid and short onset of action to facilitate dose titration to the desired effect [101,104,106,107]. These drugs should induce sedation reliably and, if necessary, a state of reduced consciousness with minimal side effects [101,104,106,107].

With regard to pharmacological therapy, a multiplicity of drugs can be used in the practice of palliative sedation [101,104,106,107]. The most frequently used drugs are benzodiazepines (midazolam, diazepam, lorazepam), neuroleptics (haloperidol, chlorpromazine, levomepromazine), barbiturates (phenobarbital), and other drugs such as anesthetics (propofol, ketamine, dexmedetomidine) (Figure 3) [101,104,106,107].

One of the novelties in this area is the use of dexmedetomidine (DEX) for symptom control and sedation. DEX is approved for use in the adult population for anesthesia care during mechanical ventilation [108]. However, clinical experience supports the application of this drug in palliative care [108]. DEX can not only induce sedation with a low risk of delirium but also minimize the side effects of other drugs [108]. DEX has a different mechanism of action, providing a type of sedation that is distinct from that induced by barbiturates, benzodiazepines, and propofol [108]. With less delirium, DEX also controls anxiety and agitation, allowing patients to remain oriented and conscious, maximizing their ability to interact with loved ones, which is a primary goal in end-of-life care [108].

According to the literature, DEX can be used in the context of palliative care, particularly in cases of hyperactive delirium or agitation; severe opioid-refractory pain; severe opioid- and benzodiazepine-refractory dyspnea; and dyskinesia, nausea, insomnia, shivering, and sedation [109]. The use of this drug should be avoided in patients with severe cardiac comorbidities. In the case of hepatic and renal failure, dose reductions are necessary [109].

## 4. Quality of Life

Assessing the quality of life (QoL) of patients in palliative care is essential [110]. The main goal of palliative care is to provide patients with the best possible quality of life [110]. The WHO defines quality of life as “an individual’s perception of their position in life in the context of the culture and value systems in which they live and in relation to their goals, expectations, standards and concerns” [110].

This concept highlights six fundamental domains: physical well-being, psychological well-being, functional autonomy, relational quality, environment, and personal beliefs or spirituality [110].

“Health-related quality of life” is a concept used to assess how an individual’s health affects how they feel physically, socially, and emotionally [111]. It is a subjective concept, as different individuals with the same disease and severity have different perceptions of their quality of life [112].

The need to assess the quality of life in cancer patients is currently undisputable, and it is no longer appropriate for the success of cancer treatment to be limited to response rates, survival time, disease-free interval, or mortality rate [113].

Cicely Saunders pioneered the modern palliative care and hospice movement by creating the first hospice and launching the concept of “Total Pain” or suffering [114,115]. Some authors studied how caregivers of patients defined quality of life or a “good death”. They concluded that this concept depends on each individual’s unique experiences but that it encompasses several dimensions, such as physical, psychosocial, and spiritual [115].

According to Higgison and Carr, measuring quality of life in oncology and palliative care is appropriate in several scenarios, such as identifying and prioritizing problems, ensuring effective communication with the patient and family, preventing any symptoms that may arise, engaging in shared decision-making, and monitoring possible changes in health and response to the treatments instituted [116].

Bernard et al. conducted a study that showed that meaning in life, especially relevant to social interactions and spirituality, had a great impact on the quality of life of patients in palliative care [117]. This study is in line with the most recent literature that shows that, at the end of life, the aspects that most contribute to the quality of life are the presence of family, relationships, and spirituality or religion, leaving the physical aspects in second place [117].

Measures to assess quality of life in palliative care are, for the most part, multidimensional [118,119].

There are currently several instruments that can be used to measure quality of life in palliative care [118,119]. The most commonly administered questionnaires were the European Organization for Research and Treatment of Cancer Quality of Life questionnaire (EORTC-QLQ), Short Form Health Survey 36 (SF-36), McGill Quality of Life Questionnaire (MQOL), and European Quality of Life Five Dimensions Questionnaire (EUROQOL EQ-5D) [118,119]. Conrad et al. found that the simple question “How satisfied are you currently with your physique and emotional well-being?’’ had the same validity as the Functional Assessment of Oncology Therapy—General scales (FACT-G) and the Palliative Outcomes Scale (POS) in the assessment of quality of life [120]. In other studies, simple questionnaires such as the Edmonton Symptom Assessment Scale (ESAS) and Palliative Outcome Scale (POS) are widely used due to their ease of implementation and frequent use in the routines of palliative care teams.

One of the goals of palliative care is to provide patients with the best possible quality of life, although it is true that information on the self-perception of quality of life at this stage of the disease is still scarce. The difficulty in interpreting and collecting data obtained through the various measurement instruments used contributes to this.

Although there is much talk about quality of life in palliative care due to its usefulness in evaluating the results of care provided to patients and their families, the truth is that this topic remains understudied [121]. Most studies do not address quality of life as a primary objective [121]. The best studies that assessed health-related quality of life in end-of-life patients were those carried out with patients followed only by palliative care teams, compared to control groups, in which patients received palliative anti-neoplastic treatment, whose main focus was to prolong life with clear advantages for the former [121].

End-of-life patients who have some type of religious belief have a better quality of life, as well as a lower risk of depression or suicide [122].

Evidence shows that the early integration of palliative care into the trajectory of chronic diseases improves the quality of life of patients, especially from a psychological and functional point of view.

## 5. Future Directions

Global policies have sought to strengthen access to palliative care [123]. However, their implementation has been difficult [123].

One of the major challenges for the future is the integration and financing of palliative care as part of general/usual clinical care [123]. According to the literature, the main future challenges for palliative care include universal access to medications such as opioids; access to universal coverage for this care; the integration of palliative care into general/usual clinical care; the development of skills among health professionals; the promotion of shared decision-making; the development of home care; and the implementation of palliative care based on the best evidence and the assessment of aspects that are significant for patients and their families [123].

More research on home-based palliative care needs to be performed in the near future. Home-based care can improve overall health outcomes and reduce healthcare costs by employing personalized quality metrics, expanding the range of services available outside the hospital, tailoring approaches to the patient and family, and improving quality of life [124]. End-of-life care, particularly in situations with high symptom burden, appears to be challenging in the home setting, requiring additional staff and teams, as well as further research to identify these challenges [124].

Regarding symptom control, it seems to us that the role of cannabinoids in cancer symptom control (particularly in advanced cancer) needs more evidence and will be the subject of more studies in the near future.

Currently, in Portugal, Law No. 33/2018, of July 18, establishes the legal framework for the use of cannabis-based drugs for medicinal purposes [125]. The use of cannabinoids has been approved for the following pathologies/symptoms: appetite stimulation in palliative care (AIDS or cancer patients undergoing treatment); chronic pain (associated with cancer or the nervous system); Gilles de la Tourette syndrome; nausea and vomiting (resulting from cancer, HIV, or hepatitis C therapy); spasticity due to multiple sclerosis or spinal cord injuries; epilepsy and severe seizure disorders in childhood; and therapy-resistant glaucoma [126].

According to the American Society of Clinical Oncology (ASCO), the use of cannabis in cancer patients is indicated for the treatment of chemotherapy-induced nausea and vomiting, total symptom burden in adults with advanced cancer, cancer pain, sleep (in patients with chronic cancer pain), low weight or poor appetite, and anxiety/depression. In addition, its use is recommended to improve quality of life [127].

## 6. Conclusions

Palliative care aims to reduce the suffering of patients, their families, and their significant others, providing them with a quality of life that would not be possible otherwise. Proper assessment and appropriate interventions to control symptoms are fundamental aspects of providing palliative care.

The multifactorial nature of these patients’ symptoms requires an initial evaluation and the monitoring of their evolution, including their intensity, impact on activities of daily living, emotional impact, and likelihood of being controlled.

Palliative care provides specific treatments for certain symptoms, which should be practical and non-invasive, including non-pharmacological and prophylactic treatments. After implementing a therapeutic plan, it is essential to adjust the medication, as undesirable side effects may occur and/or the progression of the disease may lead to the exacerbation of symptoms or the emergence of new ones.

Clinical procedures in palliative care are the same as other clinical practices, but the objective is the well-being and comfort of the patient, prioritizing the best possible quality of life.

## Figures and Tables

**Figure 1 curroncol-32-00433-f001:**
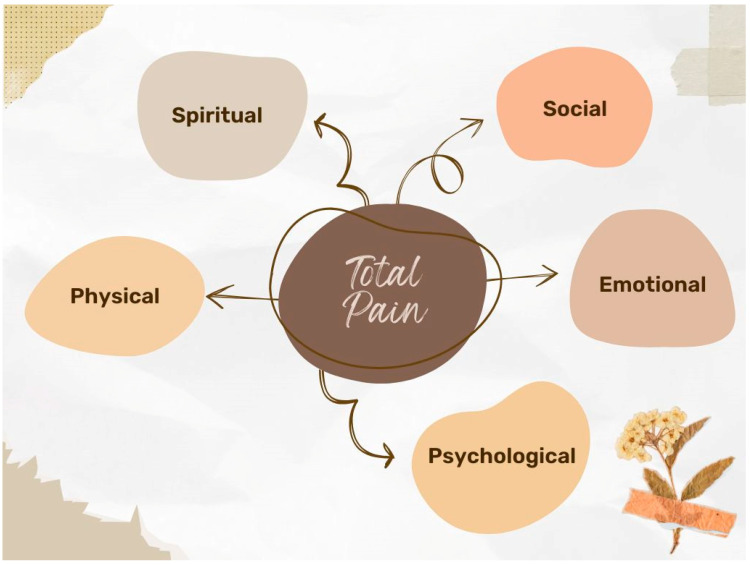
Total Pain dimensions according to Dame Cicely Saunders.

**Figure 3 curroncol-32-00433-f003:**
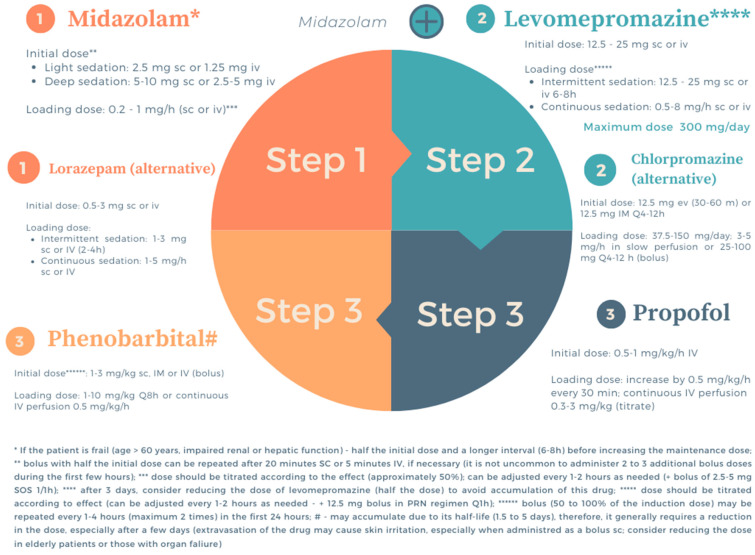
Drugs and pharmacological combinations used in palliative sedation. Based on the Palliative Sedation Protocol (Hospice and Palliative Care Federation of Massachusetts 2004 [106]), the Waterloo Wellington Palliative Sedation Therapy Protocol (Waterloo Wellington Interdisciplinary HPC Education Committee—PST Task Force 2022 [107]), Inter-professional Palliative Symptom Management Guidelines—Refractory symptoms and palliative sedation (BC Centre for Palliative Care 2017 [104]), and 2023 EAPC Guidelines [101].

**Table 1 curroncol-32-00433-t001:** Antiemetic prescribing guide based on etiology.

Nausea/Vomiting Etiology	Mechanisms Implicated	Receptors Involved	Main Drug Options
Chemical/metabolic	Stimulation of chemoreceptor trigger zone	Dopamine type 2 receptor (D2) Serotonin type 3 receptor (5HT3) Neurokinin 1 receptors (NK1)	Dopamine antagonists: Chlorpromazine; Domperidone; Haloperidol; Levomepromazine; Metoclopramide; Promethazine. Serotonin type 3 and 4 antagonists: Ondansetron; Granisetron; Cisapride. Neurokinin antagonists: Aprepitant. Acetylcholine (muscarinic) antagonists: Butylscopolamine; Scopolamine. Histamine antagonists: Cyclizine; Dimenhydrinate; Meclizine; Promethazine. Corticosteroids could be used for cranial causes: Dexamethasone; Prednisolone. Other agents: Benzodiazepines (used in nausea and vomiting associated with anxiety); Octreotide (malignant bowel obstruction); Cannabinoids.
Drugs	Stimulation of chemoreceptor trigger zone (dopamine type 2 receptor)		
	Delayed gastrointestinal transit (e.g., opioids)		
Impaired gastric emptying/ gastric stasis	Gastroparesis	Dopamine type 2 receptor (D2) Serotonin type 4 receptor (5HT4)	
Visceral/serosal causes of delayed gastrointestinal transit	Malignant bowel obstruction (stimulation of chemoreceptor trigger zone and/or stimulation of peripheral pathways)	Dopamine type 2 receptor (D2) Acetylcholine receptor (Ach) Serotonin type 3 receptor (5HT3)	
Cranial causes	Meningeal mechanoreceptor activation secondary to meningeal irritation with or without increased intracranial pressure	Histamine type 1 receptor (H1)	
Vestibular causes	Stimulation via vestibulocochlear nerve (muscarinic acetylcholine receptors, histamine type 1)	Acetylcholine receptor (Ach) Histamine type 1 receptor (H1)	
Cortical causes		Gamma-aminobutyric acid receptor (GABA) Histamine type 1 receptor (H1)	

Note: Prescribe the most appropriate first-line antiemetic; review at least every 24 h and consider parenteral administration if oral absorption is uncertain. Based on Henson et al. [70].

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
