# Peer review of "A Review on the Management of Symptoms in Patients with Incurable Cancer"

_curroncol, 2025, doi:10.3390/curroncol32080433_

Round 1
Reviewer 1 Report (Previous Reviewer 1)
Comments and Suggestions for Authors
Thank you to the authors for the revisions. The manuscript has been improved. Below are a few additional comments:
- Title: Please change "incurable cancer patients" to "patients with incurable cancer" to reflect the use of patient-first language.
- Methodology: Please change "terminally ill cancer patients" to "patients with incurable cancer".
- References: Number 74 includes 2 separate papers.
Author Response
Thank you to the authors for the revisions. The manuscript has been improved. Below are a few additional comments:
- Title: Please change "incurable cancer patients" to "patients with incurable cancer" to reflect the use of patient-first language.
Answer: In response to these comments, they decided to describe and correct the manuscript, according the suggestions.
New title - A Review on the Management of Symptoms in Patient with Incurable Cancer
- Methodology: Please change "terminally ill cancer patients" to "patients with incurable cancer”.
Answer: In response to these comments, they decided to describe and correct the manuscript, according the suggestions.
- Methodology
In this work, the authors intend to present a narrative review article on approaches to managing the main symptoms of patients in oncology and palliative care while also emphasizing the importance of their impact on quality of life.
This article presents a brief description of the main symptoms in advanced cancer, aimed at oncologists, palliative care specialists and other healthcare professionals who deal with patients with incurable cancer.
- References: Number 74 includes 2 separate papers.
Answer: In response to these comments, they decided to describe and correct the manuscript, according the suggestions.
74 - Öztürk MA. An overview of the latest ASCO recommendations about antiemetic prophylaxis for treatment-related nausea and vomiting. J Oncol Sci. 2017 undefined;3(3):99-101. doi:10.1016/j.jons.2017.09.001
Reviewer 2 Report (Previous Reviewer 2)
Comments and Suggestions for Authors
thanks for providing additional details on your reviewed manuscript
Author Response
Answer: The authors would like to thank you for taking the time to review the article in question.
Reviewer 3 Report (Previous Reviewer 3)
Comments and Suggestions for Authors
Dear Authors,
To obtain reliable and reproducible results, the review process should be systematic, encompassing specific steps and methodology. A bibliographic review does not add any knowledge to the evidence already available.
Author Response
The authors would like to respectfully point out that Reviewer 3.
The authors would also like to emphasize that the present manuscript has undergone substantial revision and improvement, both in structure and content, since the initial submission. In particular, we have clarified the scope, reinforced the rationale for the narrative review format (which is accepted by Current Oncology), and improved clarity and depth throughout.
Reviewer 3's main criticism appears to stem from a preference for a systematic review, which, while valid as an academic perspective, may not align with the intent or format of our submission. We respectfully note that the journal explicitly accepts narrative reviews, and that our work follows established methodological guidance for this type of article.
We hope that the other reviewers’ more positive evaluations, and the substantial improvements made, may also be taken into account when assessing the revised manuscript.
This manuscript is a resubmission of an earlier submission. The following is a list of the peer review reports and author responses from that submission.
Round 1
Reviewer 1 Report
Comments and Suggestions for Authors
I would like to acknowledge the significant effort made by the authors in undertaking an ambitious review of management of multiple symptoms in patients with advanced cancer. Synthesizing such a large body of literature with sufficient detail within a single paper is challenging. The usefulness of the review would be strengthened by addressing the following:
- Population of interest: The title states that the review pertains to management of symptoms in "terminally ill" patients with cancer. The descriptor insufficiently precise. Does "terminally ill" mean at any point in the trajectory of incurable cancer, or limited to the final months of life?
- Search methodology: It is unclear how the literature search that informs this review was conducted, and process used to select the referenced papers. For example, were systematic reviews/meta-analyses/guidelines prioritized? Without this information, it is difficult for the reader to judge the strength of the evidence underlying the various recommendations. For example, although opioids are recommended for dyspnea in patients with cancer, the evidence is generally considered to be of low quality (see Hui D et al, J Clin Oncol 2021).
- Currency of references: Some of the references are relatively old (for example, reference 20 for the prevalence of cancer pain is from 2006, but a more suitable one is by van den Beuken-van Everdingen MH, J Pain Symptom Manage 2016). Recent methodologically rigorous symptom management guidelines (e.g., from ASCO) have not been cited.
- Acknowledgement of the changing cancer trajectory: Although the authors correctly state in the introduction that "Care objectives should be defined on the basis of the known disease trajectory", they do not acknowledge the fact that patients with incurable cancer are living longer, which influences the approach to symptom management. This is particularly relevant to pain, for which there is a now an ICD 11 classification for chronic cancer-related pain (>3 months, see Bennett MI et al, Pain 2019) and reconsideration of the role of opioids (see Paice JA et al, J Clin Oncol 2016).
- Other comments:
- The Edmonton Symptom Assessment System is said to document intensity in "the last 24 hours". The original version of the tool (and some revised versions) use the timeframe of "now".
- It is stated that "Recently, increasing importance has been given to the concept of "Total Pain". In fact, the concept has been foundational to our understanding of pain in palliative care for decades.
- It is stated that "correct identification (of pain classification) is essential in analgesic guidance". It would be helpful to mention the NeuPSIG grading system for neuropathic pain (see Bennett MI et al, Pain 2019).
- The Distress Thermometer is incorrectly referred to as the "emotional thermometer".
- It is unclear why four paragraphs are dedicated to dexmedetomidine, for which evidence of usefulness for palliative sedation is still preliminary.
- It is stated that "The best studies that evaluated health-related quality of life in end-of-life patients were those carried out with patients followed only by palliative care teams compared to control groups, in which they only focused on curative treatments with clear advantages for the former." The control groups were actually on palliative-intent cancer treatment, i.e., life-prolonging but not curative.
Reviewer 2 Report
Comments and Suggestions for Authors
End-of-life is mentioned many times so a clear definition of both major concepts i.e. including palliative care will be appropriate.
Also see comments inside the manuscript as a clear description of what you mean by advanced cancer
this work is very interesting and relevant for practice as you did focus on the main symptoms.

Reviewer 3 Report
Comments and Suggestions for Authors
This manuscript does not have the structure of a scientific methodological paper. This manuscript is a background article, and in particular, a bibliographical research. No methodology was described, and no systematic process was adopted to guarantee the reliability of results.